# Receptor Mediated Effects of Advanced Glycation End Products (AGEs) on Innate and Adaptative Immunity: Relevance for Food Allergy

**DOI:** 10.3390/nu14020371

**Published:** 2022-01-15

**Authors:** Daniela Briceno Noriega, Hannah E. Zenker, Cresci-Anne Croes, Arifa Ewaz, Janneke Ruinemans-Koerts, Huub F. J. Savelkoul, R. J. Joost van Neerven, Malgorzata Teodorowicz

**Affiliations:** 1Cell Biology and Immunology, Wageningen University and Research Centre, De Elst 1, 6708 WD Wageningen, The Netherlands; daniela.bricenonoriega@wur.nl (D.B.N.); cresci-anne.croes@wur.nl (C.-A.C.); arifa.ewaz@wur.nl (A.E.); janneke.ruinemans-koerts@wur.nl (J.R.-K.); huub.savelkoul@wur.nl (H.F.J.S.); gosia.teodorowicz@wur.nl (M.T.); 2Food Chemistry, Department of Chemistry and Pharmacy, Friedrich-Alexander-Universität Erlangen-Nürnberg (FAU), 91054 Erlangen, Germany; hannah.zenker@fau.de; 3Department of Clinical Chemistry and Hematology, Rijnstate Hospital, 6815 AD Arnhem, The Netherlands

**Keywords:** advanced glycation end products (AGEs), immunogenicity, glycation, dAGEs, innate immunity, adaptative immunity, AGE receptors, (food) allergy, RAGE

## Abstract

As of late, evidence has been emerging that the Maillard reaction (MR, also referred to as glycation) affects the structure and function of food proteins. MR induces the conformational and chemical modification of food proteins, not only on the level of IgG/IgE recognition, but also by increasing the interaction and recognition of these modified proteins by antigen-presenting cells (APCs). This affects their biological properties, including digestibility, bioavailability, immunogenicity, and ultimately their allergenicity. APCs possess various receptors that recognize glycation structures, which include receptor for advanced glycation end products (RAGE), scavenger receptors (SRs), galectin-3 and CD36. Through these receptors, glycation structures may influence the recognition, uptake and antigen-processing of food allergens by dendritic cells (DCs) and monocytes. This may lead to enhanced cytokine production and maturation of DCs, and may also induce adaptive immune responses to the antigens/allergens as a result of antigen uptake, processing and presentation to T cells. Here, we aim to review the current literature on the immunogenicity of AGEs originating from food (exogenous or dietary AGEs) in relation to AGEs that are formed within the body (endogenous AGEs), their interactions with receptors present on immune cells, and their effects on the activation of the innate as well as the adaptive immune system. Finally, we review the clinical relevance of AGEs in food allergies.

## 1. Introduction

In the past 30 years the consumption of ultra-processed foods has almost tripled from 11% to 32% of daily energy intake, with this increased consumption of ultra-processed food being associated with a higher risk of increased mortality from numerous kinds of non-communicable diseases [1]. This diet pattern, mainly associated with Western countries, has a led to increased exposure to advanced glycation end-products (AGEs). AGEs are a heterogenous group of compounds formed as products of the Maillard reaction (MR). The MR is a non-enzymatic browning reaction, or glycation, that occurs between reducing sugars and a free amino acid group of proteins, peptides or free amino acids [2,3,4]. The MR occurs when proteins are heated in the presence of reducing sugars, e.g., during the thermal processing of food, plus it is often used in the food industry since it generates taste, color, and aroma. Additionally, it causes a number of modifications on the level of the protein structure [5]. Thus, glycation, widely used in the food industry to improve color and flavors, can change the aggregation behavior of proteins [6,7]. Processed food contains considerable amounts of AGEs (called dietary or exogenous AGEs); however, AGEs are also produced and accumulated within the body (endogenous AGEs) [8,9].

Endogenous AGEs have been shown to cause structural and functional protein alterations, as well as oxidative stress inflammation, and therefore can have pathological implications [9]. AGE accumulation can increase oxidative stress, which in turn can induce cellular dysfunction and apoptosis, ultimately leading to tissue or organ injury, therefore accelerating pathophysiological conditions [9]. Therefore, it has been postulated that both endogenous and exogenous AGEs can have a negative impact on health; nonetheless, it has to be noted that the evidence for the negative health effects of dAGEs is still lacking, as many aspects such as bioavailability, metabolism, tissue distribution and accumulation as well as the effect on immune functions are not clear for the majority of them [10]. One of the well-documented immunomodulatory mechanisms of AGEs is their ability to interact with receptors expressed by antigen presenting cells (APCs); among the other mechanisms is the receptor for advanced glycation end products (RAGE). The interaction of endogenous AGEs with RAGE can lead to the production of pro-inflammatory cytokines and has been linked to the development and progression of a number of diseases, such as type 2 diabetes, cardiovascular diseases, some forms of cancer, neurodegenerative diseases such as Alzheimer’s and Parkinson’s, liver diseases and osteoporosis [9,11,12,13], and more recently to the development of food allergies [14,15,16].

As the MR also promotes protein aggregation by means of inter- or intra-molecular crosslinking between lysine and arginine residues, glycation can initiate structural changes in proteins, resulting in the formation of ligands for receptors recognizing AGEs [17]. Moreover, the hydrophobicity, charge, aggregation and exposure of β-sheet structures are also altered due to MR, and have been suggested to participate in the immunomodulatory function of heated proteins [16,18,19]. Therefore, due to the complexity of the MR, the protein-dependent and condition-dependent differences in the obtained MRPs, as well as the variety of models applied in in vitro studies, the modes of action of dAGEs on innate and adaptative immunity described so far are not only diverse but are currently still not completely understood.

This review provides a comprehensive summary of the literature focusing on dietary AGEs (dAGEs) and how they exert their immunomodulatory effects via specific receptors on cells of the immune system. This review pays special attention to the molecular and chemical basis of the binding of glycated and heat-treated proteins to cellular receptors in order to establish a structure–function relationship. Moreover, as a consequence of the receptor-mediated immune response, the effect of dAGEs on the activation of the innate as well as the adaptive immune system is discussed. Finally, the clinical relevance of dAGEs in food allergies is reviewed and the evidence that dAGEs may modulate innate and adaptative immunity is summarized.

## 2. Maillard Reaction Products (MRPs)

### 2.1. Formation and Structural Changes in Proteins

Maillard reaction products are components formed upon heat treatment applied during food processing [20]. High heat and a prolonged cooking time of food that contains sugars increase the amount of AGEs in the food, and are accompanied by the structural changes in the food proteins [21,22]. Therefore, MR and the heat-induced structural changes in proteins take place in parallel, which makes it difficult to provide a clear distinction between the effect of heating and an effect of MR on the protein structure, e.g., when considering the aggregation [21]. For MRP formation, the initial step of heating is for the protein to start denaturing and unfolding, which is partly reversible; next comes irreversible aggregation when temperatures further increase. The extent to which each individual protein will unfold and aggregate depends on the temperature plus the time of heating, pH, protein concentration, presence of other proteins and the stability of the native structure of the proteins [23]. This protein unfolding can also lead to the exposure of structural elements located in the interior, such as the β-sheet structures, and thus can change the hydrophobicity [7]. Moreover, neo β-sheet structures can be formed upon heating via amyloid fibril formation. In parallel to these processes, the MR can promote protein aggregation and affect the types of aggregates that are formed (as shown in Figure 1A). Therefore, the effect of the heat treatment on the protein structure is a combination of an effect of the temperature itself and the interaction of amino acids with sugars due to the MR, as illustrated on Figure 1.

The MR occurs in a series of stages which can be roughly divided into the early, intermediate and final stage, all of which are mainly categorized by the products formed at each stage [24]. In brief, in the early stage of the MR, the carbonyl compound reacts with the amino compounds to form an unstable Schiff base; this process is reversible, and the so-called Schiff base subsequently rearranges to various amino-1-deoxy-ketose derivatives, which are called Amadori products (as shown in Figure 1B). Upon acidic hydrolysis, the Amadori product of lysine reacts to furosine, often used as a marker for the early stage of the MR [17]. In the intermediate stage, highly reactive α-dicarbonyl compounds are formed from the Amadori products via enolization and dehydration. Lastly, in the final stage, the dicarbonyl compounds can either react with the amino groups of the amino acids, peptides and proteins or rearrange to form AGEs [5,17,24]. The type of AGE produced highly depends on the type of sugar used in the reaction and on the processing conditions, such as temperature, time of heating, pH, and water activity, plus AGEs are bound to proteins with high contents of lysine and arginine units [25]. AGEs can be linear, e.g., Nε-carboxymethyl-lysine (CML), or crosslinked, e.g., pentosidine, where two side chains of the same proteins or different proteins are linked with each other (as shown in Figure 1C). Well-characterized AGEs in food include: CML, Nε-carboxyethyl-lysine (CEL), pyralline, pentosidine and methylglyoxal-derived hydroimidazolones (MG-H1/H2) [5,17]. The MR has also been described to have an effect on the digestibility, immunogenicity and allergenicity of food proteins; therefore, monitoring its impact on health and its use in the food industry is of great importance for the general public [16,26].

### 2.2. Endogenous AGEs: Formation and Structure

Endogenous AGE formation is part of the normal consequence of metabolism; nonetheless, this can turn pathogenic if high levels start to accumulate in tissues and the circulation [2]. Endogenous AGEs can be formed in all tissues and body fluids within the human body when the carbonyl group of reducing sugars reacts non-enzymatically with the free amino groups on proteins [27]. Even though the structures of many AGEs that are formed in vivo have not been completely determined yet, the knowledge on AGEs produced in the human body has dramatically increased in the last few decades [2]. Presently, four types of processes in the formation of AGEs under physiological conditions have been identified: (i) monosaccharide autoxidation or auto-oxidative glycosylation, (ii) Schiff’s base fragmentation, (iii) fructosamine degradation and (iv) the direct reaction of α,β-dicarbonyl compounds [2]. Endogenous AGEs include structures such as pyrraline, CEL, pentosidine and CML [28]. All of these structures were also detected in food [17].

Theoretically, as long as levels do not become too high, the effects of endogenously produced AGEs are limited by detoxification pathways [20]. However, the rate for in vivo AGE formation is determined by many factors, such as the nature and concentration of the substrate group, the glycating agents, the availability of catalytic compounds, the redox balance, the half-life of the proteins and the presence of inhibitors (e.g., aminoguanidine and pyridoxamine) [2]. This relationship is observed in patients diagnosed with diabetes, where the elevated levels of glucose accelerate the formation of AGEs; moreover, the age-specific receptor (RAGE) generates reactive oxygen species (ROS), activates inflammatory signaling cascades and therefore plays a key role in the pathogenesis of diabetic complications [29].

As stated previously, endogenous AGE formation is typically a slow process mainly affecting the function of long-lived proteins; however, their formation via reactive dicarbonyls such as methylglyoxal (MGO) can lead to quicker AGE formation [30,31]. Additionally, conditions of elevated stress and high circulating glucose levels occurring in diseases such as obesity and diabetes also increase endogenous AGE formation [32,33]. Presently, it is well-documented that an imbalance in the formation and accumulation of dAGEs contributes to a number of chronic inflammatory diseases, such as diabetes, uremia, and cardiovascular diseases, plus neurogenerative diseases including Alzheimer’s and multiple sclerosis [31,32,33,34,35,36,37,38]. Moreover, AGEs can cause the modification of guanine and adenine nucleotides of DNA. These nucleotides contain amino groups that are susceptible to glycation, which can lead to single strand breaks and mutations [33].

### 2.3. Bioavailability of Dietary AGES Induced by Food Processing

With the growing awareness of endogenous AGEs and their potential harmful effects in the human body, it became important to determine whether dAGEs contribute to the human AGE pool, and thus studies began to describe the metabolic pathway of dAGEs. To date, the available information regarding the absorption, distribution, metabolism and excretion of dAGEs is not entirely clear (as shown in Figure 2) [20], and it seems to be highly dependent on the AGEs’ structure and form (protein-bound vs. free) [17]. Nonetheless, animal and human studies concur that dAGEs are partially absorbed in the intestine and distributed across various tissues and organs [2,10,22]. It has been estimated that the absorption of AGEs into the circulation is approximately 10%, and up to 30% can be detected in the urine [2,20,22,39,40]. AGEs’ molecular weight (MW) has an impact on the absorption rate, with low MW AGEs being relatively quickly absorbed, bio-transformed and excreted, whilst high MW AGEs are normally absorbed more slowly, probably due to insufficient degradation by gastrointestinal enzymes [33]. Approximately 60% of the absorbed AGEs remain within the body for 72 h on average [1,2]. In animal studies after 3 days, more than half of the absorbed AGEs were located in the kidney and liver, but they were also found in the heart, lung and spleen [2,20]. In the colon, there is increasing evidence that dAGEs may be able to modify local microbiota metabolism and hence modulate gut integrity; this local colonic action playing an important component of the pro-inflammatory role of the dAGEs in the body [33]. Snelson et al. published a comprehensive review regarding the in vitro effects of dAGEs on gut microbiota composition and concluded that although glycated substrates do alter microbial composition in vitro, there is no consensus yet on which compositional changes occur. In addition, in vivo findings on whether dAGEs have a negative or positive influence on the intestinal microbiota are conflicting [33].

Nevertheless, studies of dAGEs in humans remain relatively scarce [41]. Moreover, the level of absorption as well as the reactivity in the body and metabolism of an individual AGE strongly depends on its chemical structure [33]. For MGO, an important marker for endogenous AGEs plus an AGE that is known for playing a role in diabetic complications, it has been reported that it is quickly degraded during the digestion process in the intestine (80% to 95%); hence, orally ingested MGO has no influence on the MGO level in vivo [42].

CML, which is highly abundant in food and because of the potential activity it has after ingestion, is normally used as a model of glycation products. Data show that diets high in dietary CML (dCML) lead to an elevation of the same compounds in urine in proportion to the amount ingested, meaning that at least a percentage of dCML is absorbed and taken into the circulation [40]. Therefore, preliminary data on the pharmacokinetics of dCML show that it is characterized by partial but rapid absorption and elimination [1]. However, the elimination of dCML seems to be incomplete, since excreted CML in feces and urine does not exceed 47%, which would suggest possible retention in organs and tissues [43]. In fact, it has been shown in a murine model that CML distributes to various organs after oral ingestion [44]. It is important to note that every glycation product differs chemically and metabolically, so not all results found on dCML can be extrapolated to every other glycation product [45].

Pyrraline is an advanced MRP usually identified in thermally treated food, and its content increases with prolonged food storage time [46]. Moreover, it has been implicated in the pathogenesis of ageing chronic renal failure (uremia) as well as diabetes and is related to complications such as inflammation, retinopathy and nephropathy [47]. It has been shown that peptide-bound pyrraline is completely bioavailable, meaning that it is proteolyzed and reabsorbed during digestion and then rapidly eliminated [47]. Hence, it is unlikely that pyrraline will be metabolized in the body, since it appears that 80% of the dietary pyrraline is absorbed, plus rapid kidney elimination was observed for all absorbed pyrraline [47]. Even though some dAGEs are efficiently cleared by the kidneys, dietary intake has to be considered when evaluating the possible physiological effects of individual MRPs, since several studies show that individuals that consume a high-AGE diet will demonstrate a significant increase in serum and urinary AGE levels [48,49,50]. These are just some cited examples of the bioavailability and metabolic transit of individual dAGEs and dicarbonyl compounds—more extensive reviews have been described in detail previously [10,45,51]. In summary, although the knowledge regarding dAGEs and their bioavailability has increased in recent years, there are many areas that demand more research. For example, a further clarification of the mechanisms involved in the absorption, metabolism and accumulation of dAGEs, improvements in analytic quantification of AGEs in foods, and a search for novel methods to inhibit AGE formation in foods as well as in vivo after ingestion are required [52].

## 3. Interaction of AGEs with the Immune System

The immune system not only provides protection from infection by pathogens, but also provides tolerance to exposure to harmless antigens; therefore, an impaired immune function is associated with increased susceptibility to infections and increased disease severity. Regarding AGEs, it is known that MRPs can interact with different types of immune receptors, including the receptor for advanced glycation end-products (RAGE), and the MRPs/AGEs have been shown to contribute to chronic inflammatory states with negative health consequences [53]. Two separate mechanisms by which AGEs are known to produce effects in the human body have been described: (i) structural deformation or cross-linking of body proteins and (ii) interaction with AGE receptors [54]. AGE cross-linking with body proteins depends on sugar concentration and the turnover rate, and it has mainly been related to the increased endogenous production of AGEs in diabetes and its comorbidities [20]. The other mechanism of action of AGEs is via AGE-sensitive receptors; circulating AGEs interact with AGE receptors and are capable of inducing cellular signaling downstream [55]. Thus, AGEs not only have a direct impact on proteins and the extracellular matrix, but they can also interact with specific cell surface receptors activating multiple mechanisms, including the production of ROS and the activation of the nuclear factor kappa B (NF-κB) pathway, resulting in enhanced production of pro-inflammatory cytokines, growth factors and adhesion molecules [2]. The AGE receptors are expressed in many cell types, including monocytes, macrophages, endothelial cells and adipocytes [20]. Based on the reviewed information above, it may be concluded that the dAGEs load contributes to the AGEs in the circulation and therefore in the AGE pool in vivo. Thus, dAGEs might be able to affect the immune system as well as metabolism in an analogical way to endogenous AGEs.

### 3.1. Critical Aspects of Binding of AGEs to AGE Receptors

To date, several AGE receptors have been identified, including RAGE, oligosaccharyl transferase complex protein 48 (AGER1), 80 K-H protein (AGER2) and galectin-3 (AGER3) [20,56,57,58]. Additionally, several AGE receptors have been identified that belong to the heterogenous scavenger receptor family, which include class A type I and II (SR-AI/II), class B type I (SR-BI), CD36 and lastly Toll-like receptors [59,60,61,62].

Each AGE receptor has unique ligand binding domains that recognize a particular structural element of the ligand. Ligands are recognized by the receptors via different mechanisms, and the ligand–receptor interactions are dependent on the type of receptor and the physicochemical characteristics of the ligand [63,64]. The structure of the different AGE receptors that have been studied in relation to dAGEs, their ligands, and the proposed sides on which each receptor recognizes the respective ligands are shown in Figure 3. Several interactions, such as electrostatic, hydrophilic and hydrophobic interactions, are involved in the binding of the receptors to their respective ligands [65,66]. Some AGE receptors show a common ligand motif, e.g., polyanionic for SR-AI, and therefore are more restricted in ligand recognition. Other receptors, such as galectin-3, show no consistent ligand pattern and thus are less restricted for specific ligands; therefore, a higher variety of receptor–ligand interactions is possible (as summarized in Table 1) [65,67].

By binding to AGE receptors, protein glycation may manipulate the uptake of dietary proteins by APCs as well as cellular signaling. For example, an enhanced uptake of glycated OVA by mature DCs as well as an increase in IL-6 production in the CD4(+) T cell co-cultures with AGE-OVA-loaded mature DC have been demonstrated [68]. Moreover, increased TNF-α production after the exposure of THP-1 macrophages to protein-bound AGEs but not free AGEs has been described [69]. Due to the large heterogenicity of AGEs and the diversity of potential receptor–ligand interactions described for the various AGE receptors, it has not yet been possible to define common motifs of AGEs that result in binding to AGE receptors. For example, for protein-bound CML, it has been indicated that the negative charge of CML could be a potential interaction motif for the binding to RAGE, sRAGE, galectin-3, and CD36 [70,71]. At the same time, other AGEs, such as protein-bound pyrraline, which has been shown to enhance T-cell activation via SRA-I, do not carry a negative charge [46]. This shows that still little is known about structure-related binding of many AGEs to a certain receptor and its molecular consequences.

Several studies have investigated MRPs’/AGEs’ binding to AGE receptors and glycation-induced immune responses [19,46,57,61,66,68,70,71,72,73]. In general, two different approaches were chosen to induce the formation of AGEs in the investigated model proteins: (i) incubating either below the denaturation temperature or above the denaturation temperature of the protein in the presence of a reducing sugar, and (ii) incubation with chemicals that result in a specific modification with an individual AGE. The latter method has repeatedly been shown not to affect the secondary structure of the protein [46,70], and also incubation below the denaturation temperature of the protein is considered to minimize 3D structural changes. Nevertheless, Cardoso et al. showed that even heating at moderate temperatures can result in protein aggregation [7]. This was especially promoted by the MR. It is thus important to sufficiently monitor 3D structural changes in the investigated proteins after glycation, which has so far only been performed in a few studies [19,46,66,72]. This is crucial, as recent studies pointed out that not only glycation but also other 3D structural changes such as aggregation, increased surface hydrophobicity and β-sheet formation/exposure are determinants for binding to RAGE and its soluble form (sRAGE), galectin-3, SRA-I and CD36 [19,66,70,72,73].

For example, it was shown that the binding of heated β-lactoglobulin to sRAGE differs depending on the heating conditions (dry vs. wet conditions, high vs. low temperature as well as the presence or absence of reducing sugars), and that aggregation also plays an important role in the formation of sRAGE-binding ligands. Notably, glycation was not an indispensable requirement for the formation of sRAGE ligands [19]. Likewise, the uptake of heated β-lactoglobulin by THP-1 macrophages can be explained by hydrophobicity, the exposure of β-sheets and aggregation, rather than glycation itself [70]. Moreover, aggregates with a molecular weight above 100 kDa that are formed upon the heating of β-lactoglobulin both in the absence and presence of reducing sugars were more potent ligands to galectin-3, CD36 and SRA-I in both THP-1 cells and monocyte-derived human dendritic cells than the fraction of lower molecular weight [72]. The fact that the presence of sugar was not required to induce binding to AGE receptors indicates that it is also important that alongside unheated controls, glycation controls (protein heated in the absence of reducing sugars) should be added to the sample sets when testing for AGE receptors responses. Not only is this a better control of the 3D structural changes necessary, but the qualitative and quantitative assessment of the formed AGEs is another critical point when investigating the immunomodulatory properties of AGEs. Previously, a wide variety of methods to monitor glycation in the model food proteins were used to assess binding to AGE receptors, with some providing more informative than others. Fluorescence intensity measurement as well as the OPA-assay were used, as they were easy and quick methods to determine the overall extent of glycation relative to the unheated protein; however, they do not allow for the quantification of specific AGEs [18,74,75,76]. To quantify specific AGEs, immunoassays such as ELISA have often been used; however, interference in the quantification by other substrates and optimization of the matrix is crucial for its optimal use. Additional analytical methods include reverse-phase high-performance liquid chromatography (RP-HPLC), ultraviolet light (UV), gas chromatography mass spectrometry (GC-MS), and liquid chromatography mass spectrometry (LC-MS), the latter two of which are not only targeted but are also highly sensitive and are considered as state-of-the-art methods for qualifying and quantifying individual AGEs [77]. In the search for specific AGE receptor-binding AGEs and their structural motifs that result in this binding, this will be indispensable; nevertheless, until now, these analytical methods are not comprehensively applied in this field of research. Nonetheless, a few individual protein-bound AGEs have been directly tested as ligands for specific AGE receptors such as pyrraline, CML, CEL, and MG-H1 [46,70,71,78,79]. In the most recent study, Zenker et al. attempted to discriminate the role of different protein modifications in binding to AGE receptors by testing β-lactoglobulin which was selectively modified with CML vs. β-lactoglobulin that was glycated with lactose below its denaturation temperature. Their findings show that protein-bound CML is a ligand for sRAGE, CD36, and galectin-3 as measured by inhibition ELISA and that the negative charge of CML is a determinant for the binding. [70]. In summary, AGE receptors may be responsive to structural modifications other than AGEs and thus it is crucial to properly monitor 3D structural changes, to add glycation controls and to sufficiently characterize the AGEs that are formed during the applied glycation procedures.

### 3.2. Immunogenicity of AGEs

As previously stated, AGEs have been shown to contribute to chronic states of inflammation within the human body. Inflammation is part of the body’s immune defenses, both innate and adaptative, therefore compromising a series of cellular and chemical barriers that aim to control endogenous and exogenous harmful stimuli [80]. One of the potential immunological modulatory effects of AGEs is the capacity of AGEs to be recognized by receptors on the surface of the immune cells [14,15,81]. This not only has the capacity for contributing to low-grade inflammation and local inflammation in tissues expressing RAGE but also to food allergies, since the induction of oxidative stress and enhanced cytokine expression are consequences of this binding (as shown in Figure 4) [82].

Food proteins through interaction with pathogen recognition receptors (PRR), which includes macrophage scavenger receptors and RAGE, may modulate additional downstream immunological effects promoting T cell activation and differentiation, leading to Th2 responses in food allergies [45,83]. Further potential immunological modulatory effects of AGEs include the ability to create novel epitopes that are recognized by IgE, the capacity to enhance inflammatory conditions as well as oxidative stress, and causing a reduction in the diversity of the intestinal flora which is associated with an enhanced susceptibility to allergies [17,46,53,64,66,72,81,82,83,84,85].

Regarding the immunogenicity of AGEs through receptor recognition, it is crucial to note that some receptors, such as RAGE, only bind the AGEs and produce cytokine as a result, but other receptors, such as galectin-3, also internalize the proteins, leading to enhanced antigen presentation of the protein to which the AGEs are bound, which may lead to the activation of food protein-specific T cells [72,83,86,87]. Therefore, the initiation of adaptative immunity by AGEs will occur indirectly via antigen presentation, ultimately leading to T-cell activation.

#### Influence of AGEs on the Innate and Adaptative Immune System

The innate immune system recognizes microbes or foreign objects directly through pattern recognition receptors (PRRs), including RAGE [74]. RAGE is the most researched AGE receptor, since in response to the AGEs, load is the main up-regulator of cell activation [9,28,30,56,63,64]. It is known that interactions between AGE and RAGE trigger various intracellular signaling cascades which are followed by the transcription of a range of genes which perpetuate pro-inflammatory signals [88]. More specifically, AGE–RAGE binding leads to the activation of NF-κB, a key player in the activation of pro-inflammatory pathways, including an increase in the cytokine expression, growth factors and adhesion molecules [2,3,68,89], but also the generation of ROS [55,88]. There is a body of evidence demonstrating that the interaction of RAGE with endogenous AGEs induces oxidative stress and inflammation [20], and increasing data on a similar effect of dAGEs; however, these studies are mostly based on in vitro work or experiments in rodents [19,72,73,87,89].

Current evidence suggests that RAGE shares ligands and intracellular signaling pathways such as NF-kB activation with TLRs, and can therefore cooperate in strengthening inflammatory response [62,90,91]. Moreover, studies show cross-talk between TRLs and RAGE, acting together through the recruitment of homo and hetero-oligomers that strengthen inflammatory responses [92]. Liu et al. reported that TLR4 expression increased in AGE-exposed macrophages, which was then followed by the activation of RAGE/ROS signaling; thus, in macrophages, over-expression of RAGE elevated both ROS and TRL4 expression [93]. Therefore, it is possible that RAGE/ROS/TLR4 signaling is responsible for AGE-induced macrophage polarization [93]. Macrophages have a high degree of plasticity, meaning that they can alter their function rapidly and polarize either to a more pro-inflammatory (M1) state, or an anti-inflammatory (M2) state [94,95]. Currently, evidence shows that macrophages treated with endogenous AGEs lead to polarization of macrophages into the M1 state via MAPK signaling [96,97]. Nonetheless, at the moment, as most studies are conducted with endogenous AGEs and not dAGEs, it remains unknown as to whether dAGEs are capable of inducing the same response in macrophages. AGEs can also be generated by means of interactions between oxidized lipids and proteins, thus contributing to inflammation [52], and recently Xu et al. reported that AGEs could also increase lipid accumulation in macrophages, possibly by upregulating the expression of RAGE [98]. These findings support the theory that the downregulation of RAGE expression or blocking the binding of AGEs to RAGE may provide an interesting therapeutic avenue in the management of chronic diseases such as diabetes.

In order to induce an adaptive immune response, dAGEs first need to be internalized by APCs and subsequently presented to T cells (as shown in Figure 5). This is strongly influenced by the efficiency of antigen binding, uptake and processing, as well as the activation status and production of cytokines by myeloid APC [99]. Receptor-mediated endocytosis is an efficient method of antigen uptake that makes possible the activation of adaptative immune responses at low antigen exposure and may be facilitated by galectin-3, Fc receptors, dectin-1, 2 and 3, DC-SIGN and mannose receptors [16]. Recently, more evidence has been found that dAGEs are recognized and internalized by a number of the aforementioned receptors also expressed by DCs [3,18,20,22,72,86]. Teodorowicz et al. showed the binding and internalization of heat-treated and glycated β-lactoglobulin by human monocyte-derived DCs [72]. Ge et al. reported that AGEs obtained from bovine serum albumin induced the maturation of DCs and increased their capacity to stimulate T cell proliferation and cytokine secretion [100]. Nevertheless, Price et al. showed that AGEs derived from adrenocorticotrophic hormone failed to show maturation markers of DCs and their capacity to stimulate primary T-cell proliferation [101].

By using ovalbumin (OVA), an egg white allergen, as a model, Heilmann et al. attempted to identify specific glycation structure(s) that had food allergenic potential by influencing T-cell immunogenicity in a murine study [46]. Their research was one of a few attempting to identify the AGE structures responsible for the activation of T-cell immunity using OVA modified with CEL, CML, and pyralline (Pyr). The T-cell immunogenicity of different glycated OVA was assessed by co-culturing murine OVA-specific CD4+ T-cells with bone marrow-derived DCs. OVA modified by Pyr (Pyr-OVA) showed an enhanced production of IL-2, IL-17A and IFN-y compared to native OVA, demonstrating an increased CD4+ T-cell immunogenicity. Furthermore, the scavenger receptor (SR) was involved in the uptake of Pyr-OVA by bone marrow dendritic cells (BMDCs). Therefore, this study showed that pyralline was capable of inducing enhanced allergen uptake by DCs via an association with SR class A (SR-A), and thus enhancing CD4+ T cell activation and IgE production, aiding in the understanding of how the MR enhances the potential allergenicity of food allergens [46]. Similarly, Ilchmann et al. demonstrated the uptake of glycated OVA by myeloid DC via receptor mediated endocytosis involving SR class-A type I and type II, and showed that the production of IL-4 was enhanced by OVA-specific CD4+ T-cells [86].

There are also studies showing that glycation may not always lead to an activation of T-cell immunity. For example, Perusko et al. described the immunological effect of glycated ß-lactoglobulin (BLG), demonstrating that glycation significantly increased the uptake by BMDCs via receptor-mediated endocytosis via SRs [102]. Nevertheless, despite higher degradation by lysosomal enzymes, glycated BLG demonstrated a lower ability to induce the production of Th1- and Th2-type cytokines in a co-culture of BMDC with BLG-specific CD4+ T cells [102]. In contrast, a different study investigating the immunogenicity of bovine β-lactoglobulin showed that the heat-induced formation of amyloid-like structures, aggregates and increased hydrophobicity are the features determining the binding to APCs [18]. Neither study included the effects of enzymatic digestion of the processed/glycated protein on the binding to the specific receptors. The importance of glycation in vivo was suggested by a study which showed that glycated aggregates of β-lactoglobulin are less sensitive to digestion and therefore maintain their binding capacity to RAGE and Gal-3 [72]. Finally, β-lactoglobulin modified with CML was shown to be recognized by sRAGE, galectin-3, and CD36 [70]. Recently, it has been shown that the RAGE expressed on T cells is involved in the activation of the T-cell signaling cascade and may be an important mechanism in the response of T cells to inflammatory mediators. This suggests that RAGE may also play a role in the direct activation of T cells via AGEs present in the circulation, contributing to inflammation and enhanced T-cell reactivity [83].

Hence, during the processing of an antigen, various factors such as antigen uptake, the activation of DCs, the generation of peptides, and the stability and density of the MHC peptide complex can affect the immunogenicity of the antigen [103]. Since the antigen’s fate is determined by the intracellular degradation of the antigen in APC, antigens with a higher susceptibility to endolysosomal enzymes have a weaker capacity to prime T cells [103]. Therefore, the ability of dAGEs to activate the T-cell immunity may be determined by a number of factors, including the heterogeneity of AGEs formed under different conditions as well as the unique amino acid composition of proteins determining the formation of amyloid-like structures, hydrophobic motives and aggregates, which are the features determining the binding to APCs. In conclusion, the heterogeneity of AGEs and the diversity of their receptors make it difficult to formulate unequivocal conclusions on the structure–function relationship in the activation of adaptive immunity. Thus, more well-defined and unified studies are needed in order to define the AGE structures responsible for binding to certain receptors and the consequences of this binding on the level of both innate and adaptive immunity.

## 4. Association between dAGEs and Food Allergy

IgE-mediated food allergy prevalence is rapidly increasing, particularly in Western countries, and evidence suggests immune system dysfunction in the development and persistency of food allergy [104]. Since human genetics have not been capable of radical modification in recent decades, it is more plausible that the ways in which genes function have been altered by environmental factors, influencing epigenetic processes such as methylation, ubiquitination and histone acetylation [105,106].

It has been suggested that a Western diet, typically high in AGEs, has an effect on allergenicity via T helper (Th2), including IL-4, IL-5, and IL-13, the pro-inflammatory cytokines Il-1, IL-6, and IL-8, and TNF-α and alarmins [90]. Type I allergic immune responses are primarily characterized as T-helper cell (Th2)-driven, which results in the formation of allergen-specific IgE antibodies, leading to mast cell activation upon secondary contact with the responsible allergen [44,80].

By modulating T-cell immunogenicity and the antigen presentation of food proteins by pathogen recognition receptors (PRR), which include macrophage SR and RAGE, they contribute to T-cell differentiation and Th2 responses [44,80,81]. For example, group 2 innate lymphoid cells (ICL2s) have recently emerged as important cells in the pathogenesis of allergic asthma because they were shown to be major producers of IL-5 and IL-13 in the recruitment and accumulation of ICL2s in the lung of mice in response to allergens [84]. Moreover, the levels of endogenous sRAGE increased in sputum samples of asthma patients correlated with disease severity [85]. Further potential immunological modulatory effects of AGEs include the ability to induce novel IgE binding epitopes, the capacity for enhancing inflammatory conditions and oxidative stress and causing a reduction in the diversity of the intestinal flora, which leads to an enhanced susceptibility to allergies [16,17,41,80].

The possible influence of the MR on the potential allergenicity of certain food allergens has been a topic of recent interest [14,15,16,17]. So far, studies have shown that the effects of the MR in food allergenicity are diverse, mostly dependent on the thermal stability of allergens but also on the types and concentrations of reducing sugars, food matrix composition and treatment conditions (e.g., temperature, pH, duration and moisture) (as shown in Figure 5) [17]. The affinity of allergens for specific IgE antibodies may be influenced by glycation, which in turn has an impact in the electric charge and hydrophobicity of proteins [18,19]. Moreover, glycation with AGEs induces either the masking of epitopes or the generation of neo-allergens [15]. Extracts from roasted peanuts have been shown to induce higher levels of IgE than raw peanuts, and Mueller et al. described the binding of peanut allergens that had been specifically modified by AGEs to RAGE, establishing that RAGE does interact with AGE-modified recombinant Ara h 1 but not with unmodified recombinant Ara h 1 [106]. Teodorowicz et al. described that MR-type neo-allergens in processed soy caused a strong allergic reaction in soy-sensitized individuals [107]. Evidence shows that to induce an allergic immune response, the MR-modified protein needs to be recognized and taken up by APCs and subsequently be presented to T cells (as shown in Figure 5) [86]. Moreover, several studies have identified RAGE and several other receptors as described above from the scavenger family as the main receptors recognizing glycated food proteins [46,57,58,61,72].

The false alarm hypothesis, proposed by Smith et al., suggests that the signaling of immune cells by RAGE-activated APCs is important for the role of AGEs in food allergies [108]. Moreover, it has been proposed that dAGEs might induce alarmin signaling, and thus have the potential to mimic tissue damage through glycation [108,109]. In other words, dAGEs could mimic innate alarms and skew towards allergic responses in certain subjects that have a genetic and environmental predisposition [108]. Currently, there is no direct evidence that AGEs trigger food allergies through interaction with RAGE, though RAGE is highly expressed on DCs, macrophages, T lymphocytes and B cells [2,3,22]. Thus, evidence that directly links food allergies to the AGE–RAGE axis is lacking, plus it is still not known which glycation molecules bind exactly to which receptors in vivo. We propose here that activation of APC via not only RAGE but also other receptors induces a proinflammatory environment, but that uptake and antigen presentation via AGE receptors that can internalize the food allergen with AGEs is a key mechanism linking the activation of the innate immune system by AGEs with the activation and differentiation of adaptive, food allergen-specific Th2 responses (Figure 5). So far, there is no precise evidence to predict the consequence of food-derived AGEs on the allergenicity of food proteins, and further studies are needed to understand the biological and the immunological characteristics and consequences of MPRs [3,22]. Moreover, even though the negative effects on human health of dAGEs are more widely recognized, it remains a question as to whether habitual dietary AGE intake per se is to blame [52,110]. It is important to note that the essential concept when introducing a low-AGE diet relates more to the manner of cooking and not the actual nutrient composition of the food. Food processing and certain forms of cooking/baking can hide, destroy or disclose allergenic epitopes by conformation changes or, in some cases, changing the digestibility of the protein [111]. Thus, conveying the importance of food processing/cooking and its potential to alter the allergenic properties of proteins should be of great value in populations where food allergies are on the rise [112].

## 5. Conclusions

The evidence reviewed above shows that dAGEs may act as molecules that can modulate innate and adaptative immunity, and thus contribute to low-grade inflammation, food allergies and non-communicable diseases. dAGEs may influence the activation of the immune system in two different manners—firstly, via the interaction with RAGE which does not lead to the internalization of the ligand [72] but does activate pro-inflammatory pathways as described for endogenous AGEs [113]. Further research is necessary to determine which RAGE ligands are involved in promoting RAGE-dependent responses. In vitro studies suggest that protein-bound CML might play a role in RAGE activation [70], while heat-induced protein aggregation may also play an important role [72]. Since RAGE is capable of recognizing various ligands characterized by ß-sheets and fibril formation, this could place RAGE in the group of PRRs, which are important for the activation of the innate immune response via food-derived ligands and contribute to non-communicable diseases [94,95].

Secondly, dAGEs may influence the activation of the immune system through the binding of dAGEs to the receptors RAGE, galectin-3, CD36 and SR-A, which internalize the ligands and mediate the interaction of APCs with the adaptive immune system, and which may facilitate T-cell activation and skewing, thus leading to allergic responses. Although the number of studies showing the T-cell skewing by dAGEs is limited, the enhanced uptake and presentation of food allergens with dAGEs by APCs are well-documented.

Therefore, it can be implied that the interaction of dAGEs and specific receptors on APC play an important part in the immunogenic effects of dAGEs. However, not all studies use the same types of proteins and methods of glycation and characterization of AGEs and structural changes of proteins. Furthermore, information regarding the effects of digestion on glycated proteins and their effects on APC is limited at present. As a result, no definitive conclusions on the interaction of AGEs with the immune system in vivo can be drawn at present. Future human studies are needed to elucidate the relevance of these mechanisms in health and disease.

## Figures and Tables

**Figure 1 nutrients-14-00371-f001:**
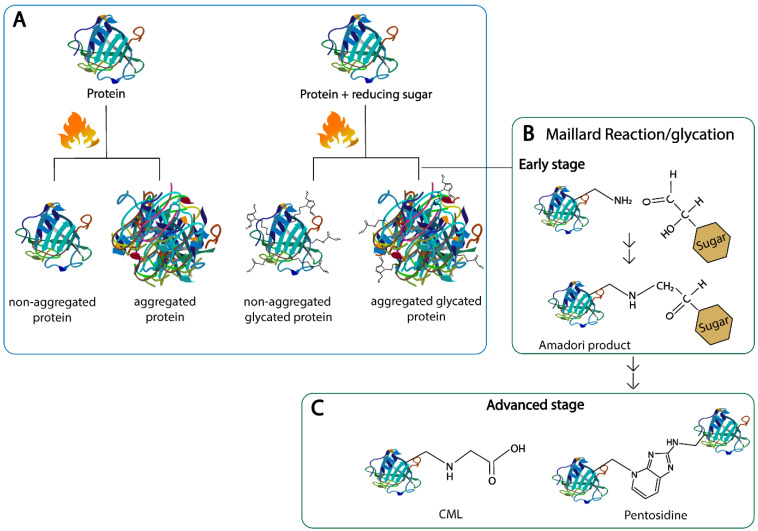
Structural changes in proteins upon heating with and without reducing sugars. (**A**): protein aggregation, (**B**): early Maillard reaction, (**C**): advanced stage of the Maillard reaction. N^ɛ^-carboxymethyl lysine (CML) and pentosidine are shown as representatives for linear and cross-linking of heterocyclic advanced glycation end products, respectively.

**Figure 2 nutrients-14-00371-f002:**
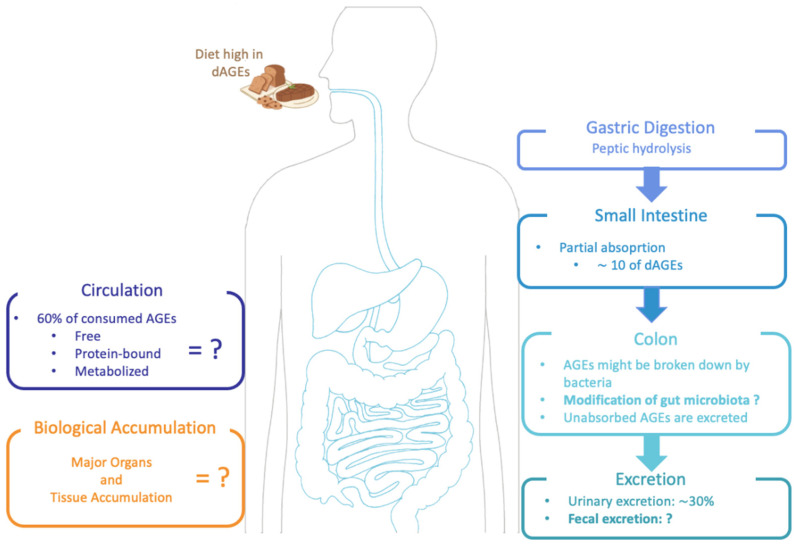
Suggested metabolic pathway of dAGEs: absorption, digestion and excretion in the human body.

**Figure 3 nutrients-14-00371-f003:**
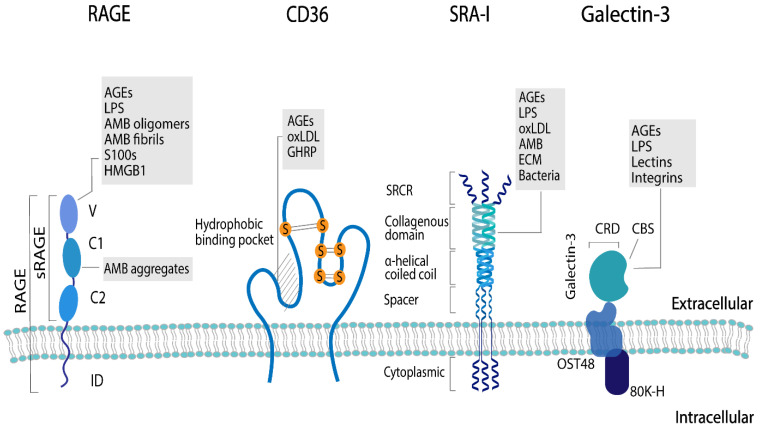
Structure and ligands of the AGE receptors mostly researched in relation to dAGEs: receptor for advanced glycation end products (RAGE), the soluble form of RAGE (sRAGE), CD36, scavenger receptor class A type I (SRA-I), and galectin-3. The ligand-binding domains of the different receptors are: V-domain (V), C1- and C2-domain (C1 and C2), and intracellular-domain (ID), scavenger receptor cysteine-rich structure (SRCR), oligosaccharyltransferase 48 complex (OST48), carbohydrate-binding site (CBS), and carbohydrate recognition domain (CRD). The ligands for the different receptors are advanced glycation end products (AGEs), amyloid-β (AMB), (ECM), growth hormone-releasing peptide (GHRP), high mobility group protein 1 (HMGB1), lipopolysaccharide (LPS), and oxidized low-density lipoprotien (oxLDL).

**Figure 4 nutrients-14-00371-f004:**
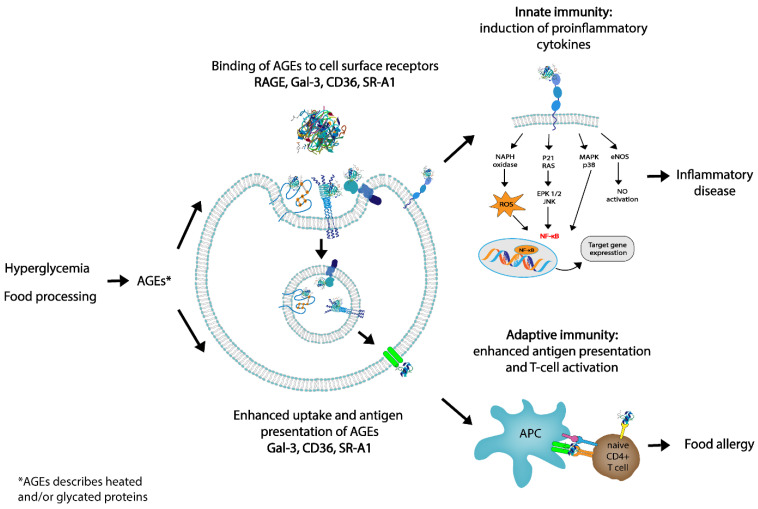
Role of dietary advanced glycation end products (AGEs) in innate and adaptive immunity. By targeting AGE receptors, dAGEs may influence innate immunity via interaction with RAGE or by binding to AGE receptors which internalize the ligands: galectin-3 (Gal-3), CD36, and scavenger receptor class A type I (SR-A1). Presentation of antigen to T cells may facilitate T-cell activation and skewing, possibly leading to allergic responses. In this way, dAGEs may contribute to both innate and adaptive immunity.

**Figure 5 nutrients-14-00371-f005:**
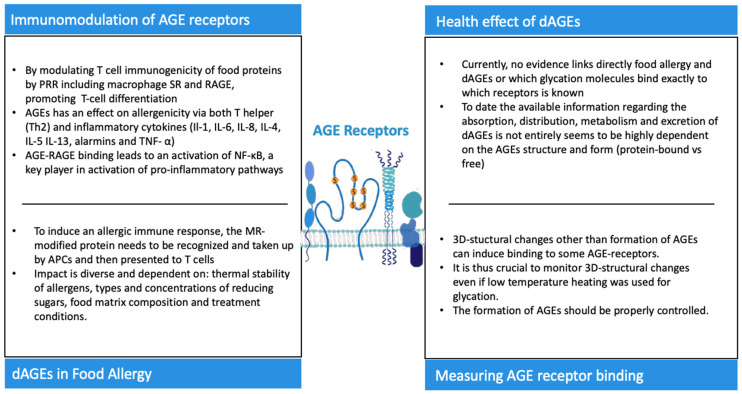
Key messages regarding AGE receptors and their role in the innate and adaptative immune system plus the health effects of dAGEs.

**Table 1 nutrients-14-00371-t001:** Structural elements of AGE receptors that have been researched in relation to dAGEs: RAGE, galectin-3, SR-A and CD36: their role in AGE binding and the forces that result in the interaction of AGEs with the respective AGE receptor.

Receptor	Structure	AGE Binding Sides	Forces for AGE Interaction
RAGE	EC: one V-type domain, two C-types domains and a short transmembrane domain	V-type domain	Electrostatic
	IC: cytoplasmic tail		
Galectin-3	Component of AGE-R complex has a carbohydrate recognition domain (CRD) and a carbohydrate binding side (CBS)	CBS	Hydrophilic interactions via hydrogen bonds, and hydrophobic interactions, specifically the CH-π interaction explains binding to lectins and lipopolysaccharides. For specific AGEs unknown.
SR-A	EC: scavenger receptor cysteine-rich structure (SRCR), collagenous domain, α-helical coiled coil, and spacer as well as an intracellular cytoplasmic	Collagenous domain	For specific AGEs unknown.All ligands are macromolecular and polyanionic. For apo-A and apo- E amphipathic α-helix suggested as a potential recognition motif. Dual cation-binding site proposed as main domain for ligand binding via SR-A, hence electrostatic interactions.
CD36	Two transmembrane domains, an EC loop with glycosylation sites and two short IC tails	Hydrophobic binding pocket located at the highly glycosylated sites	Electrostatic, via a positively charged moiety that binds to negatively charged ligands, based on studies with diacylglycerol and oxLDL as ligands.

## Data Availability

Not applicable.

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
