# Peer review of "Receptor Mediated Effects of Advanced Glycation End Products (AGEs) on Innate and Adaptative Immunity: Relevance for Food Allergy"

_nutrients, 2022, doi:10.3390/nu14020371_

Round 1

Reviewer 1 Report

I have written comments that I was going to send but 24th of November has me at a hospital with a very sick dog. The main points I would like to make are

  1. Lipid conjugation with sugars also contribute to inflammation
  2. our sugar intake has increased but fructose has increased by 700% in the last 30 years. Fructose is a fast track to MGXL.
  3. That AGEs can cause DNA damage, interfere with spindle formation and mitochondrial function. The authors clearly discuss the formation of aggregated proteins and graphics are great. 
  4. It might be uplifting to include a reference to dietary AGEs ..Jaime Uribari published this about 7 years a

Author Response

On behalf of the authors of the Manuscript ID-1526770 we would like to thank you for your comments and also hope your dog is now safe and happy back at home. 

We are attaching a word file with a point by point answer to your comments and suggestions, please keep in mind that the number lines were the new sections have been added refer to the lines in the CLEAN version of the manuscript, since the manuscript with change can distort the line count. 

Thank you very much for time and consideration.

Reviewer 2 Report

  • Please define the process for conducting a literature search in the abstract.
  • More comprehensive and recent literature review is needed. The paper would be benefit from greater consideration of the following: (1) transition metals that accelerate the MR (e.g., Fe2+); (2) Nutritional strategies for controlling of MR (e.g., polyphenols); (3) the role of RAGE in the pathogenesis of asthma/allergic airway inflammation (please refer to J Allergy Clin Immunol. 2015;136(3):747-756.e4; Paediatr Respir Rev. 2017;23:40-49); (4) the effect of calorie/dietary restriction on the AGE load; (5) the effect of AGEs on the gut microbial composition and circulating metabolites such as SCFA (Line 181-184, Ref #33); (6) the effect of high AGE diet (animal-derived foods that are high in fat and protein) in comparison with other widely known restrictive diets such as VLCKD on gut immune system and inflammation (Line 51-62; Please refer to Int J Mol Sci. 2020; 21(24): 9580). These sources may also add more value to the paper and I would suggest authors referring to (J Allergy Clin Immunol. 2010, 125(1):175-83.e1-11; Food and Agricultural Immunology. 2015, 26(6): 835-847; Food and Agricultural Immunology 2013, 24(4): 433-443; World J Immunol. 2016, 6(1): 19-38).
  • Authors need to prove a gap in the literature, and clarify the importance of this review at the end of introduction. I would suggest that the authors to present the aim of the paper with regards to what is currently known in vivo/vitro studies, thus highlighting the added value of this study.
  • My main concern is the lack of methodology applied to choice of presented papers. Authors reviewed several studies but it is not clear how these studies were chosen for inclusion in the review. The method section should include search terms, study type (vivo/vitro), inclusion/exclusion criteria, databases and a period of time encompassed by the search (starting year chose to be included in this review).
  • There are old references (e.g., Ref#39,53,56,78,84) that can be replaced with more recent ones.
  • Authors should follow the journal guidelines for references. For example, Author 1, A.B.; Author 2, C.D. Title of the article. Abbreviated Journal Name Year, Volume, page range.

Author Response

On behalf of the authors of the Manuscript ID-1526770 we would like to thank you for your comments and suggestions

We are attaching a word file with a point by point answer to your comments and suggestions, please keep in mind that the number lines were the new sections have been added refer to the lines in the CLEAN version of the manuscript, since the manuscript with change can distort the line count. 

Thank you very much for time and consideration.

Round 2

Reviewer 2 Report

No further comments.